# MEIS2 Is an Adrenergic Core Regulatory Transcription Factor Involved in Early Initiation of TH-MYCN-Driven Neuroblastoma Formation

**DOI:** 10.3390/cancers13194783

**Published:** 2021-09-24

**Authors:** Jolien De Wyn, Mark W. Zimmerman, Nina Weichert-Leahey, Carolina Nunes, Belamy B. Cheung, Brian J. Abraham, Anneleen Beckers, Pieter-Jan Volders, Bieke Decaesteker, Daniel R. Carter, Alfred Thomas Look, Katleen De Preter, Wouter Van Loocke, Glenn M. Marshall, Adam D. Durbin, Frank Speleman, Kaat Durinck

**Affiliations:** 1Department for Biomolecular Medicine, Ghent University, Medical Research Building (MRB1), Corneel Heymanslaan 10, B-9000 Ghent, Belgium; jolien.dewyn@ugent.be (J.D.W.); carolina.decarvalhonunes@ugent.be (C.N.); anneleen.beckers@gmail.com (A.B.); pieterjan.volders@ugent.be (P.-J.V.); bieke.decaesteker@ugent.be (B.D.); katleen.depreter@ugent.be (K.D.P.); wouter.vanloocke@ugent.be (W.V.L.); franki.speleman@ugent.be (F.S.); 2Department of Pediatric Oncology, Dana-Farber Cancer Institute, Harvard Medical School, Boston, MA 02215, USA; markw_zimmerman@dfci.harvard.edu (M.W.Z.); ninaw_leahey@dfci.harvard.edu (N.W.-L.); thomas_look@dfci.harvard.edu (A.T.L.); 3Lowy Cancer Research Centre, Children’s Cancer Institute Australia for Medical Research, UNSW Sydney, Randwick, NSW 2031, Australia; bcheung@ccia.org.au (B.B.C.); dcarter@ccia.org.au (D.R.C.); glenn.marshall@health.nsw.gov.au (G.M.M.); 4School of Women’s and Children’s Health, UNSW Sydney, Randwick, NSW 2031, Australia; 5Department of Computational Biology, St. Jude Children’s Research Hospital, Memphis, TN 38105-3678, USA; brian.abraham@stjude.org; 6School of Biomedical Engineering, University of Technology Sydney, Ultimo, NSW 2007, Australia; 7Kids Cancer Centre, Sydney Children’s Hospital, Randwick, NSW 2031, Australia; 8Department of Oncology, Division of Molecular Oncology, St. Jude Children’s Research Hospital, Memphis, TN 38105-3678, USA; adam.durbin@stjude.org

**Keywords:** neuroblastoma, mouse model, MYCN, transcriptome analysis

## Abstract

**Simple Summary:**

Neuroblastoma is a pediatric tumor originating from the sympathetic nervous system responsible for 10–15% of all childhood cancer deaths. Half of all neuroblastoma patients present with high-risk disease, of which nearly 50% relapse and die of their disease. In addition, standard therapies cause serious lifelong side effects and increased risk for secondary tumors. Further research is crucial to better understand the molecular basis of neuroblastomas and to identify novel druggable targets. Neuroblastoma tumorigenesis has to this end been modeled in both mice and zebrafish. Here, we present a detailed dissection of the gene expression patterns that underlie tumor formation in the murine TH-MYCN-driven neuroblastoma model. We identified key factors that are putatively important for neuroblastoma tumor initiation versus tumor progression, pinpointed crucial regulators of the observed expression patterns during neuroblastoma development and scrutinized which factors could be innovative and vulnerable nodes for therapeutic intervention.

**Abstract:**

Roughly half of all high-risk neuroblastoma patients present with MYCN amplification. The molecular consequences of MYCN overexpression in this aggressive pediatric tumor have been studied for decades, but thus far, our understanding of the early initiating steps of MYCN-driven tumor formation is still enigmatic. We performed a detailed transcriptome landscaping during murine TH-MYCN-driven neuroblastoma tumor formation at different time points. The neuroblastoma dependency factor MEIS2, together with ASCL1, was identified as a candidate tumor-initiating factor and shown to be a novel core regulatory circuit member in adrenergic neuroblastomas. Of further interest, we found a KEOPS complex member (*gm6890*), implicated in homologous double-strand break repair and telomere maintenance, to be strongly upregulated during tumor formation, as well as the checkpoint adaptor Claspin (*CLSPN*) and three chromosome 17q loci *CBX2*, *GJC1* and *LIMD2*. Finally, cross-species master regulator analysis identified FOXM1, together with additional hubs controlling transcriptome profiles of MYCN-driven neuroblastoma. In conclusion, time-resolved transcriptome analysis of early hyperplastic lesions and full-blown MYCN-driven neuroblastomas yielded novel components implicated in both tumor initiation and maintenance, providing putative novel drug targets for MYCN-driven neuroblastoma.

## 1. Introduction

Neuroblastoma is a deadly childhood cancer arising from sympatho-adrenergic nervous progenitor cells [1]. The high-risk neuroblastoma genomic landscape typically presents with a low mutational burden, while highly recurrent patterns of large segmental chromosomal imbalances and focal gains and losses are observed in most cases [2,3,4,5]. The driver role of MYCN in neuroblastoma initiation has been supported through murine and zebrafish models overexpressing MYCN in the developing sympathetic neuronal lineage [6,7,8]. In this study, we performed gene expression profiling of different stages of tumor development in the TH-MYCN mouse model. The TH-MYCN mouse model represents a highly penetrant transgenic model of neuroblastoma that expresses high levels of human MYCN in the developing sympathetic tissues using a conditional rat tyrosine hydroxylase promoter [9]. Histological examination of developing TH-MYCN^+/+^ sympathetic ganglia showed that tumor initiation is characterized by pre-cancerous neuroblast hyperplasia in the first 2 weeks of life, followed by clonal selection of malignant neuroblasts and progression to tumor formation by 6 weeks of age [10,11,12]. A number of contributing factors have been identified in TH-MYCN tumor initiation such as p53 deregulation, metabolic adaptation, micro-RNA deregulation and the MYCN binding proteome, many of which represent promising therapeutic targets in MYCN-driven neuroblastoma [10,13,14,15,16,17,18,19,20,21]. Nevertheless, determinants of MYCN-driven initiation remain unclear, in particular the key events that allow progression from a premalignant hyperplastic phenotype to frank malignancy. To explore the role of MYCN in early tumor development in more depth, we performed a unique combined cross-species and time-course transcriptome analysis of early hyperplastic lesions and full-blown tumors in the TH-MYCN murine model, together with expression data from primary human neuroblastoma and mouse normal sympathetic neuronal development. We identified MEIS2 as an early tumor initiation associated factor, as well as multiple novel MYCN-upregulated dependency genes. 

## 2. Materials and Methods

### 2.1. Publicly Available Datasets Used for Analysis

In this study, several publicly available datasets were used: (1) gene expression profiles from 649 neuroblastoma tumors (GSE45547) to perform survival analysis, (2) gene expression profiles from 498 primary neuroblastomas (GSE49711) for scoring signatures of murine MYCN-driven neuroblastoma in a cohort of primary human neuroblastomas as well as master regulator analysis, (3) ChIP-seq data for GATA3, ISL1, HAND2, PHOX2B and H3K27ac in SK-N-BE(2)-C and N206 (Kelly, DSMZ, Leibniz Institute, Braunschweig, Germany) neuroblastoma cells (GSE94822) and (4) expression data from a cohort of primary medulloblastoma cases (GSE21140, =103).

### 2.2. TH-MYCN Neuroblastoma Progression Model

Animal experiments were performed according to the Guide for the Care and Use of Laboratory Animals. Personnel that carried out the described experiments received appropriate training in animal care and handling. TH-MYCN^+/+^ mice were sacrificed at days 7 (*n* = 4) and 14 (*n* = 4) after birth to harvest sympathetic ganglia containing foci of neuroblast hyperplasia and at week 6 of life to harvest advanced neuroblastoma tumors (*n* = 4) [11]. Additionally, we dissected the same sympathetic ganglia from wild-type mice at day 7 (*n* = 4), day 14 (*n* = 4) and week 6 (*n* = 4) of life to control for mRNA expression changes during normal development.

### 2.3. RNA Isolation and Gene Expression Analysis of TH-MYCN^+/+^ and Wild-Type Samples

Murine total RNA was isolated using the miRNeasy Mini Kit (Qiagen, Hilden, Germany, 217004) according to the manufacturer’s instructions. The samples were profiled on Agilent SurePrint G3 Gene Expression Microarrays, Santa Clara, CA, USA) according to the manufacturer’s protocol. Data were summarized and normalized with the vsn method in the R statistical programming language using the limma package (R-package limma, Roswell Park Comprehensive Cancer Center, Buffalo, NY, USA). Probes with a log_2_ expression of less than 4 in more than 20/24 samples were considered not expressed and filtered out. Additionally, for each unique gene symbol, only the probe with the most significant differential expression between TH-MYCN^+/+^ and wild-type samples (two-way ANOVA, *p*-value of interaction term) was retained for further analysis. Differential expression analysis was performed using the R-package limma, as described in the user guide for time-course experiments. Statistical testing was performed using the empirical Bayes quasi-likelihood F-test. Microarray profiling results for all samples are available in the ArrayExpress database under Accession Number E-MTAB-3247. Furthermore, the processed data can be visualized via the R2 microarray Analysis and Visualization Platform (http://r2.amc.nl (accessed on 22 September 2021) [22]) under experiment “Exp Nb Hyperplasia TH-MYCN—Ghent—24—custom—agmge8 × 60”. Furthermore, it can also be visualized through the Shiny application (https://shiny.dev.cmgg.be/app/01_hyperplasia_time_series (accessed on 22 September 2021) [23]) that we developed for this study. Gene signature scores were calculated using a rank-scoring algorithm, and two-way ANOVA was applied to assess the interaction between the genotype and the age. For gene ontology analysis, EnrichR, (https://maayanlab.cloud/Enrichr/ (accessed on 22 September 2021) [24]) was used with the default settings to identify enriched functional classes [25].

### 2.4. MEIS2 ChIP-Sequencing

In total, 100 × 10^6^ SK-N-BE(2)-C or N206 (Kelly) neuroblastoma cells were processed for ChIP-sequencing as previously described [26,27,28]. Briefly, pellets were prepared by fixing cells in 1% formaldehyde for 15 min and quenching in 1 M Tris pH 7.5. Nuclei were prepared using the Sigma Nuclei Isolation Kit (Sigma Aldrich, Saint Louis, MO, USA (#NUC-101)) and sonicated using a Branson digital sonifier (Emerson, Saint Louis, MO, USA) as previously. Antibodies to MEIS2 were from Sigma Aldrich (Sigma Aldrich, Saint Louis, MO, USA, HPA003256). ChIP-seq libraries were prepared using the KAPA hyperprep ChIP library kit (Roche, Basel, Switzerland) following the manufacturer’s settings and were sequenced on an Illumina Nextseq 500 machine (Illumina, San Diego, CA, USA). Raw sequencing data were mapped to the human reference genome (GRCh37/h19). Peak calling was performed using MACS 1.4 [29]. Normalization of the data was performed for the library sizes by multiplying each value by (1,000,000/(total read count)).

The raw data files for all samples are available in the ArrayExpress database under Accession Number E-MTAB-10620.

### 2.5. MEIS2 Pharmacological Inhibition

Human neuroblastoma cell lines SK-N-BE(2)C, IMR32 and N206 (Kelly) were grown in RPMI1640 medium supplemented with 10% fetal calf serum, 100 IU/mL penicillin/streptomycin and 2 mM L-glutamine. All cell lines were cultured in 5% CO_2_ atmosphere at 37 °C. Meisi-2 was obtained from Meinox Pharma Technologies (Istanbul, Turkey) [30]. In order to determine the dose–response curve of Meisi-2, cells were seeded in a 384-well plate at a density of 10^3^ − 3.5 × 10^3^ cells per well, depending on the cell line. Cells were allowed to adhere overnight, after which these were exposed to Meisi-2. The treatment was performed using the D300 TECAN instrument, (Tecan Group Ltd, Männedorf, Switzerland). Cell proliferation was monitored for 72 h, in which pictures were taken through IncuCyte^®^ Live Cell Imaging System (Essen Bioscience, Newark, UK). Each image was analyzed through the IncuCyte^®^ Software. Cell proliferation was monitored by analyzing the occupied area (% confluence) of cell images over time. The experiment was repeated three times for each cell line in order to obtain independent biological replicates. The dose–response curve, different inhibitory concentration values and area under the curve (AUC) were computed through GraphPad Prism version 7.00 (GraphPad Software, San Diego, CA, USA). The analysis was performed through the ECanything equation, assuming a standard slope of −1.0.

### 2.6. Human Neuroblastoma Regulatory Network Assembly and Master Regulator Analysis

A neuroblastoma specific network model of transcriptional regulation was built using ARACNe (Algorithm = “Adaptive Partitioning”, Mode = “Complete”, Data Processing Inequality (DPI) Tolerance = 0, Resampling = 100) based on 498 primary neuroblastoma samples (GSE49711) and a list of 2213 transcription related genes [31]. Next, the enrichment of each regulon of this network on the TH-MYCN mouse-derived neuroblastoma gene expression signature described above was inferred by the VIPER algorithm [32], as implemented in the VIPER package for R available from Bioconductor (Roswell Park Comprehensive Cancer Center, Buffalo, NY, USA).

### 2.7. Overall Survival Analysis in Neuroblastoma Tumors

Overall survival analyses were performed on 649 neuroblastoma tumors for which patient survival (*n* = 476) data were available. Curves were generated using R2: Genomics Analysis and Visualization platform using the Tumor Neuroblastoma-Kocak-649-custom-ag44kcwolf dataset. Patients were divided into two groups using median expression as a cut-off.

### 2.8. Statistical Analysis

The appropriate statistical analysis and the according tests are indicated in each of the respective figure legends. Statistical tests were performed with the R software using a Student *t*-test (*p* < 0.05 was considered to be statistically significant) or a two-way ANOVA (analysis of variance) test to assess the variability between groups. Kaplan–Meier analysis with log-rank statistics was used for survival analysis.

### 2.9. Gene Set Enrichment Analysis and Cytoscape Network Visualization

Gene set enrichment analysis was performed using the C5_BP MSigDB collection, (Broad Institute, Main, MA, USA). To visualize the enrichment results, the Cytoscape plugin “Enrichment Map” was used. The FDR-q cut-off value is given in the figure legends.

## 3. Results

### 3.1. Validation of Dynamic Gene Regulation during Murine TH-MYCN Tumor Development Using Established Neuroblastoma Gene Signatures

Using principal component analysis, the global variation across the obtained gene expression profiles (Figure 1A) was assessed. The samples from the first week after birth of both genotypes (TH-MYCN^+/+^ and wild type) clustered together, implying that the overall changes in the transcriptomes of this set of samples were comparable with the average variability between biological replicates and therefore used as a reference set for further downstream analyses.

At later time points (2 and 6 weeks after birth), the overall changes in the transcriptome profiles increased to the average variability between replicates and both genotypes cluster further apart, suggesting strong transcriptional differences between neuroblastoma tumors and normal ganglia.

In the next step, we evaluated the dynamic behavior of previously reported genes implicated in neuroblastoma biology. Based on a literature survey, we selected a total of 140 genes that were previously reported as oncogenes, codrivers or tumor suppressor genes in neuroblastoma and evaluated their differential expression between TH-MYCN^+/+^ versus wild-type mice at 6 weeks after birth. We found *ALK*, *ASCL1* and *CHAF1A* amongst the top upregulated genes and *DKK1* (a previously reported MYCN downregulated gene), *DKK3* [33] (a known miR-17-92 target), *RET* [21] (known to be downregulated together with other adrenergic and cholinergic markers) and *CHD5* [34] amongst the top downregulated genes (Figure 1B).

Furthermore, expression of known MYC(N) transcriptional target genes, summarized as a MYCN activity score [35,36], significantly increased over time in the TH-MYCN^+/+^ samples compared to wild type for the tested MYCN signatures (Figure 1C). This observation is in line with the hypothesis that induction and sustained MYCN activity in the sympathetic ganglia of TH-MYCN^+/+^ mice gives rise to the occurrence of neuroblastoma tumors in these anatomic locations [11]. Furthermore, genes involved in late neuronal sympathetic differentiation [35], also summarized in an activity score, decreased significantly over time in TH-MYCN^+/+^ (Figure 1D), showing that late neuronal sympathetic differentiation markers are suppressed during murine neuroblastoma development.

For the activity scores of adrenergic- and mesenchymal-specific gene signatures [37], the adrenergic signature significantly increased over time in TH-MYCN^+/+^ (Figure 1E). In contrast, the mesenchymal gene signature was significant for the genotype or time component separately (respectively, *p* = 0.006 and 0.006), but not significant for genotype–time interaction (*p* = 0.12, see Figure 1E), underscoring the adrenergic cell state of the hyperplastic lesions and full-blown tumors.

Moreover, TH-MYCN^+/+^ mice display strong upregulation of immature adrenergic markers during tumor development in contrast to mature adrenergic marker genes [38] (Appendix A, respectively).

Recent analysis also revealed a plethora of key MYC(N) interaction partners [39]. We interrogated this core set of defined interactors in the dynamic TH-MYCN transcriptome dataset and showed a strong differential regulation over the course of hyperplastic lesion and full-blown neuroblastoma tumor formation for most interactors (Figure 1F), including the PLK1 kinase, which has a crucial role in MYC protein stabilization [40], and TOP2A that was recently shown to control RNA polymerase II pause release in a dynamic interplay with Aurora kinase A [41].

### 3.2. Upregulated Expression of ASCL1 and MEIS2 as Candidate Early Initiating Events in Murine TH-MYCN Tumor Development

Following the validation of the murine TH-MYCN temporal transcriptome dataset, we next performed a total of four different transcriptome analyses, marked as Analysis A, B, C and D as depicted (Figure 2).

For the first analysis (Figure 2A), we sought candidate genes strongly regulated during tumor initiation with differential expression only between weeks 1 and 2 (not yet at week 6). To this end, we identified 92 differentially expressed genes in TH-MYCN^+/+^ versus wild-type samples uniquely at 2 weeks versus week 1 (absolute log_2_ fold change threshold of 1 and adjusted *p* < 0.01, (Figure 3A, top). We also scored these signatures in a large cohort of primary neuroblastoma tumor samples (GSE120572). This analysis showed that the signatures from hyperplastic lesions in the TH-MYCN^+/+^ model do not yet recapitulate the transcriptome profiles from primary neuroblastomas as expected (Figure 3A, bottom).

Pathway enrichment for this set of genes was performed by EnrichR, revealing, amongst others, a role for genes implicated in embryonic stem cell pluripotency pathways (Figure 3B, *red bars*), while, amongst others, glycosaminoglycan degradation and TGF-β signaling were shown to be reduced (Figure 3B, *blue bars*). Gene set enrichment analysis for TH-MYCN^+/+^ versus wild-type samples at 2 weeks (not yet at 6 weeks) of age versus week 1 resulted in significantly enriched gene sets upregulated in TH-MYCN^+/+^ versus wild type (Figure 3B, right).

Next, we specifically screened this list for transcription factors that could cooperate with MYCN in the earliest step of neuroblast transformation and identified Nhlh2 and Meis2 as possible candidates. First, *Nhlh2* (alias *Hen2*), is a neural developmental transcription factor [42] that has been reported to bind and cooperate with LMO3 to repress *HES1*, which itself is a negative regulator of ASCL1 [43]. ASCL1, together with PHOX2B, is a known critical developmental factor in sympathetic and parasympathetic nervous system development and has been reported as a canonical factor of the adrenergic core regulatory circuitry in neuroblastoma co-regulated by MYCN and LMO1 [44], but its specific oncogenic dependency role in this adrenergic circuitry is elusive. Second, *MEIS2* was previously reported as a key factor supporting neuroblastoma cell survival and established as a neuroblastoma-specific dependency factor by DEPMAP consortium screening [45,46]. Further, DNA binding motif analysis suggested a role as a core regulatory circuit member in MYCN-driven neuroblastoma [47], but so far, ChIP-seq data were not available to support this. To further investigate this, we performed landscaping of the MEIS2 genome-wide DNA binding sites in SK-N-BE(2)-C and N206 (Kelly) neuroblastoma cells by ChIP-sequencing. We observed a strong global overlap of MEIS2 binding sites to those of MYCN, PHOX2B, GATA3, HAND2, ISL1 as well as H3K27ac signal in the same cell line (shown for SK-N-BE(2)-C, Figure 3C). Moreover, we could show that all these factors cobind each other encoding genomic locus, as exemplified for the *MEIS2* and *PHOX2B* locus (Figure 3D). In addition, MEIS2 also co-binds various consensus-defined, super-enhancer-marked loci in MYCN-amplified cell lines [28], such as *RGS5* [48] (Figure 3D).

In order to gain further insight into the possible early role of Ascl1 and Meis2 in MYCN-driven tumor formation compared to the other core regulatory circuit members, we compared their time-resolved expression during murine TH-MYCN-driven neuroblastoma tumor development. Remarkably, Hand2, Phox2b, Gata3 and Isl1 were highly expressed in both TH-MYCN^+/+^ early hyperplasia and wild-type ganglia at week 1 and remained at this level in both tissues throughout further tumor development. In contrast, Ascl1 and Meis2 showed transcriptional upregulation from week 1 to week 2 in TH-MYCN^+/+^, with declining expression in wild type ganglia at this time frame, suggesting they execute a distinct role during tumor formation and possible also in their maintenance of the core regulatory circuit network (Figure 3E).

Recent single-cell transcriptome analysis of the developing sympatho-adrenal lineage evidenced [49] strong enriched MEIS2 expression in the population of sympathoblast cells that resemble the neuroblastoma cancer cell phenotype compared to chromaffin and Schwann cell precursor cells (Figure 3F, left). We also verified the MEIS2 expression pattern in the recent single-cell transcriptome data of the adrenal medulla by Jansky et al. (Figure 3F, right) [50], showing prominent expression in the neuroblast population from which neuroblastoma cells are assumed to originate.

Last, we evaluated MEIS2 as a novel putative druggable target in neuroblastoma by exposing IMR32, N206 (Kelly) and SK-N-BE(2)-C neuroblastoma cells to a range of concentrations of a recently developed inhibitor Meisi-2 (Meinox Pharma Technologies, Istanbul, Turkey), targeting both MEIS1 and MEIS2 and recently described to modulate hematopoietic stem cell activity [30]. We could show that pharmacological MEIS2 inhibition negatively impacted on neuroblastoma cell confluence (Figure 3G).

### 3.3. KEOPS Protein Complex Member LAGE3 Expression Is Strongly Upregulated towards Full-Blown Murine TH-MYCN Tumor Formation

For our second analysis (Figure 2B), we aimed to identify further factors interacting with MYCN overexpression in the developing sympathetic lineage and fully established tumors. To this end, we compared transcriptomes from week 1 versus 2 and 6 and identified a total of 377 significantly differentially expressed genes in TH-MYCN^+/+^ versus wild-type mice (absolute log_2_ fold change threshold of 1 and adjusted *p* < 0.01, top-100 shown in Figure 4A). The strongest upregulated gene in the latter analysis is *gm6890*, a homolog of the human *LAGE3* gene and encodes for a protein that is part of the strongly evolutionary conserved “Kinase, Endopeptidase and Other Proteins of Small size” (KEOPS) complex implicated in t6A tRNA biosynthesis [51], homologous double-strand break repair and telomere maintenance [52,53]. We also scored these signatures in a large cohort of primary neuroblastoma tumor samples (GSE120572). This analysis showed that the signatures from weeks 2 and 6 after birth in the TH-MYCN+/+ model significantly recapitulate the transcriptome profiles from primary neuroblastomas (Figure 4A, bottom).

From the four other core KEOPS protein complex encoding components (*c14orf142*, *TP53RK*, *TPRKB* and *OSGEP*), *Osgep* is also significant differentially expressed during murine TH-MYCN tumor development compared to wild type (Appendix A), while high expression of all components, except *OSGEP*, was significantly correlated to poor overall patient survival (GSE45547, *n* = 649) (Appendix A).

Other strongly upregulated genes in this set were *Dlk1*, highly expressed in early sympathetic neuron progenitors [54] and *Plk1*, a known therapeutic target in neuroblastoma [55]. Gene ontology analysis on the set of protein-coding genes revealed among the upregulated genes an enrichment for cell cycle and DNA repair factors (Figure 4B, left, *red bars*), while acetylcholine synthesis was shown to be downregulated, a neurotransmitter that is synthesized in some primary neuroblastomas (Figure 4B, left, *blue bars*). Gene set enrichment analysis for TH-MYCN^+/+^ versus wild-type samples at 2 and 6 weeks of age versus week 1 resulted in significantly enriched gene sets upregulated in TH-MYCN^+/+^ versus wild type (Figure 4B, right).

### 3.4. Cell Cycle and DNA Repair Genes Are Upregulated Late during TH-MYCN-Driven Tumor Formation

For our third analysis (Figure 2C), we considered genes upregulated after week 6 only and revealed 2638 differentially expressed genes (absolute log_2_ fold change threshold of 1 and adjusted *p* < 0.01, top-100 shown in Figure 4C, top) that were not yet significantly differentially expressed at 2 weeks of age. We found, amongst others, strong upregulation of *Pbk*, a kinase that was recently identified as a key LIN28B target driving the proliferation and self-renewal of neuronal stem cells [56] as well as *Bcl11a*, previously shown to be overexpressed in high-risk neuroblastoma [57]. Several of the top downregulated genes are *bona fide* or presumed tumor suppressor genes, including *Dkk1* and *Dkk3* (Wnt signaling antagonists) [33,58]. Further, *Acyp2* was previously described to induce differentiation of neuroblastoma cells [59]. We also scored these signatures in a large cohort of primary neuroblastoma tumor samples (GSE120572). This analysis showed that the signatures from week 6 after birth (full-blown tumors) in the TH-MYCN^+/+^ model significantly recapitulate the transcriptome profiles from primary neuroblastomas (Figure 4C, bottom).

Gene ontology enrichment on the set of protein-coding genes indicated a clear upregulation of genes implicated in cell cycle control and DNA damage repair (Figure 4D, red bars), while cAMP-dependent signaling was downregulated, which would otherwise drive neuroblastoma cell differentiation (Figure 4D, blue bars) [60]. Gene set enrichment analysis for TH-MYCN^+/+^ versus wild-type samples at uniquely 6 weeks of age versus week 1 did not result in any significantly enriched gene set.

### 3.5. MEIS2 and Other MYCN-Driven Dependency Genes Are Upregulated during Murine TH-MYCN-Driven Neuroblastoma Tumor Formation

To further delineate putative relevant novel codrivers in MYCN overexpressing neuroblastomas, we performed cross-section of dynamically upregulated genes in TH-MYCN mice from Weeks 1 to 2 with the “CRISPR Avana screen” identified dependencies from the DEPMAP initiative common between included MYCN-amplified and neuroblastoma cell lines [28,61]. This resulted in the identification of *Meis2* as a single overlapping gene (Figure 5A, *left*).

This is in keeping with its identification as an exclusively differentially expressed factor from Weeks 1 to 2 in TH-MYCN^+/+^ versus wild-type mice (Figure 3A). Next, the same analysis was performed for genes uniquely differentially expressed between mice 1 and 6 weeks of age (Figure 4C). In total, 24 overlapping genes could be retrieved, with high expression of 15 of those significantly correlating to poor overall patient survival in human primary neuroblastoma (Figure 5A, left, GSE45547, *n* = 649). Single-cell RNA-seq data of the sympathoadrenal lineage and adrenal medulla respectively indicate that *CDCA8* displays nearly exclusive strong expression in the proliferating sympathoblasts [49] and cycling neuroblast population [50] (Figure 5B).

Three of those genes (*CBX2*, *GJC1*, *LIMD2*) are encoded on 17q, a chromosomal region invariably gained in high-risk neuroblastoma (Figure 5C). CBX2, a core component of the PRC1 complex, was previously shown to inhibit neurite development [62]. Further, LIMD2 was recently identified as a miR-34a target in lung cancer [63], a putative tumor suppressor miRNA in neuroblastoma [64].

*SOX11*, the locus which maps to the short arm of chromosome 2, which is often gained together with MYCN amplification in high-risk neuroblastoma, is a key oncogenic factor in multiple cancer entities including mantle cell lymphoma [65]. Furthermore, it is functionally connected to both early and late steps during neurogenesis, and we recently established a putative crucial role in adrenergic neuroblastoma cells for regulation of chromatin accessibility and cell identity [66].

Last, when screening for MYCN co-dependencies that were differentially expressed between mice of both 2 and 6 weeks of age compared to week 1 (Figure 4A), four overlapping genes could be retrieved, with high expression of three of those significantly correlating to poor overall patient survival in human primary neuroblastoma (Figure 5D, top, GSE45547, *n* = 649).

From the CRC network, only *ASCL1* emerged from this analysis, given its different dynamic expression during murine TH-MYCN-driven tumor formation compared to HAND2, PHOX2B, GATA3 and ISL1 (see Figure 3E). In addition to *ALK* and *ASCL1* that were previously reported in the context of neuroblastoma, a novel interesting candidate gene “Claspin” (*Clspn*) was identified, for which recently a critical role was described in protection of cancer cells from replication stress and which has been reported as a key gene in normal developmental controlled gene amplification in *Drosophila* [67]. Single-cell RNA-seq data of the sympathoadrenal lineage and adrenal medulla respectively indicate that CLSPN displays nearly exclusive strong expression in the proliferating sympathoblasts [49] and cycling neuroblast population [50] (Figure 5D, bottom) We also evaluated the overlap of the set of significantly upregulated genes in the course of murine TH-MYCN-driven neuroblastoma tumor development with the neuroblastoma specific dependencies as defined by the recent pediatric cancer dependency map from Dharia et al. [68]. When again only considering those genes that also significantly correlate with a poor overall neuroblastoma patient survival, we identified 11 additional candidate dependency genes that were significantly differentially expressed (absolute log_2_ fold change threshold of 1 and adjusted *p* < 0.01) between TH-MYCN^+/+^ and wild-type mice exclusively at Week 6 compared to Week 1 (*AGBL5*, *ATAD5*, *CHEK1*, *HDX*, *NEDD1*, *PABPC1*, *PHF19*, *POLD1*, *PYCR1*, *SMC2* and *WNT5B*) that were not identified as “MYCN-amplified/neuroblastoma” specific dependencies by the Avana CRISPR screen from the Depmap initiative. Altogether, the generated transcriptome dataset accurately recapitulates gene expression patterns of human MYCN-driven neuroblastoma while allowing selecting novel target genes for future functional studies.

### 3.6. Cross-Species Master Regulator Analysis in MYCN-Driven Neuroblastoma

Previously, Rajbhandari et al. reported the identification of TEAD4 as a novel master regulator in MYCN-amplified neuroblastoma based on transcriptome analysis of independent primary human neuroblastoma cohorts [69]. In a last step of our data mining effort, we took advantage of our established gene signatures of TH-MYCN^+/+^ versus wild-type mice 6 weeks after birth together with previously validated bioinformatics pipelines to infer small regulatory modules driven by master regulator transcription factors that could uncover novel tumor-specific vulnerabilities for innovative therapeutic approaches (Figure 2D). To this end, a human neuroblastoma-specific transcriptional network was built with ARACNe, an unbiased algorithm that infers direct transcriptional interactions, based on gene expression profiles [70]. This analysis was based on the gene expression in a large heterogeneous cohort of primary neuroblastoma tumors (GSE49711, *n* = 498) to interrogate regulatory connectivity of 2213 genes involved in transcriptional regulation based on iRegulon [31]. Following the establishment of a murine neuroblastoma signature and a human neuroblastoma-specific transcriptional network, the algorithm “Virtual Inference of Protein activity by Enriched Regulon Analysis” (VIPER) [32] was used to prioritize potential master regulators in MYCN-driven neuroblastoma.

The resulting data were then further filtered for de facto transcription factors and resulted in a ranked master regulator hit list (top 50 is shown in Figure 6A), ranked by the “normalized enrichment score” (NES) for each of the identified regulons.

Our ranking order, however, does not account for the differential expression of these respective putative master regulators in the course of TH-MYCN-driven neuroblastoma tumor formation. Therefore, we reranked the top-50 identified putative master regulators according to the extent of upregulated expression from Weeks 1 to 6 in TH-MYCN^+/+^ versus wild-type mice and filtered out the zinc finger (ZNF)-type proteins given their ambiguity of mouse orthologs and displayed the top-20 in Figure 6B (panel 1). This consists of, amongst others, HMGB2, the DREAM complex members FOXM1, MYBL2 and E2F8, DEPDC1, the epigenetic regulators DNMT1, DNMT3A and EZH2, the c-MYB proto-oncogene and the neurodevelopmental transcription factor TCF3 [42]. The DREAM complex supports a plethora of essential biological functions, including cell proliferation, G2/M transition, DNA damage repair and tissue homeostasis [71]. To this end, we compiled a DREAM/FOXM1 signature [72] and could show significant upregulation during TH-MYCN-driven neuroblastoma formation (Figure 6B, panel 2). We verified the expression of *HMGB2* in single-cell RNA-seq data of the developing sympathoadrenal lineage, showing prominent expression in the proliferating sympathoblasts [49]. This was in line with the strongest *HMGB2* expression in the cycling neuroblast cell population of the adrenal medulla [50] (Figure 6B, panel 3).

## 4. Discussion

We present for the first time a detailed dissection of a time-resolved protein-coding transcriptome dataset generated through dissection of hyperplastic lesions and full-blown tumors from the murine TH-MYCN neuroblastoma model. First, we identified MEIS2 as a putative key transcriptional driver during early neuroblastoma tumor formation. ChIP-sequencing revealed MEIS2 as a component of the previously established adrenergic core regulatory transcription factor circuitry. Interestingly, both MEIS2 and ASCL1 revealed a pattern of dynamic regulation during murine TH-MYCN-driven neuroblastoma tumor formation that clearly deviated from those observed for other canonical CRC members (HAND2, PHOX2B, GATA3 and ISL1), pointing towards a putative distinct role for MEIS2 and ASCL1 in the core regulatory circuit network during tumor formation. Interestingly, ASCL1 has been previously shown to act as a pioneering factor in fibroblasts [73] and could thus fulfill a similar role in the context of neuroblastoma tumorigenesis. *Further in vivo* functional analysis is warranted to unravel the precise role of MEIS2 during early neuroblastoma tumorigenesis. We also identified *Gm6890*, a human ortholog of LAGE3 and a core component of the KEOPS complex, amongst the top upregulated genes across the analyzed time frame of tumor development. The main subunits of this complex display a strong evolutionary sequence conservation and play a functional role in telomere maintenance (supports uncapping and elongation), transcription, chromosomal segregation [53,74,75] and DNA repair [52]. Our analysis hints towards a putative co-dependency role for this protein complex and warrants further investigation. Identification of genetic (co)-dependencies is key in preclinical research, as those genes are essential for the fitness of the cancer cell under study and point towards valuable entry points for therapeutic intervention, well-illustrated with the identification of the BCR-ABL fusion kinase in chronic myeloid leukemia that could be specifically blocked by imatinib [76]. Such dependencies were recently broadly inferred in the context of pediatric solid malignancies [68]. To derive further novel and functionally relevant putative MYCN co-dependencies during neuroblastoma tumor progression, we also integrated the time-resolved transcriptome dataset with common hits of CRISPR based screening data in MYCN-amplified and neuroblastoma cells [28,61] and with neuroblastoma specific hits as defined by the pediatric cancer dependency map initiative [68]. This analysis again highlighted our finding of MEIS2 as a putative key early dependency during murine TH-MYCN-driven neuroblastoma. In addition, other putative interesting candidates include amongst others CBX2 (a core PRC1 subunit required for binding to H3K27me3 marked domains), CDCA8 (a subunit of the “Chromosomal Passenger complex” (CPC) and direct MYCN target gene contributing to the aggressive phenotype of MYCN-amplified neuroblastoma [77]), Claspin (a key mediator of the ATR-CHK1 pathway), and PIM1 (a key driver of ALK inhibitor resistance in neuroblastoma [78]). Last, our study defined for the first time through a cross-species integrative transcriptomics approach putative master regulators for MYCN-driven neuroblastoma tumor development and revealed FOXM1 and the DREAM complex members MYBL2 and E2F8 as top-scoring candidates. Notably, MEIS2 is a direct transcriptional regulator of FOXM1 and indirect regulator of DREAM expression [46], warranting further functional studies to further dissect the mechanistic role of MEIS2 during neuroblastoma tumor development.

## 5. Conclusions

In conclusion, we present an in-depth characterization of the dynamic transcriptome profiles of TH-MYCN-driven murine hyperplastic lesions and established tumors and the integration with both primary human neuroblastoma tumor expression data, epigenetic and functional genomics data to identify and validate candidate cooperating dependencies, suitable as putative candidates for a precision medicine approach in neuroblastoma.

## Figures and Tables

**Figure 1 cancers-13-04783-f001:**
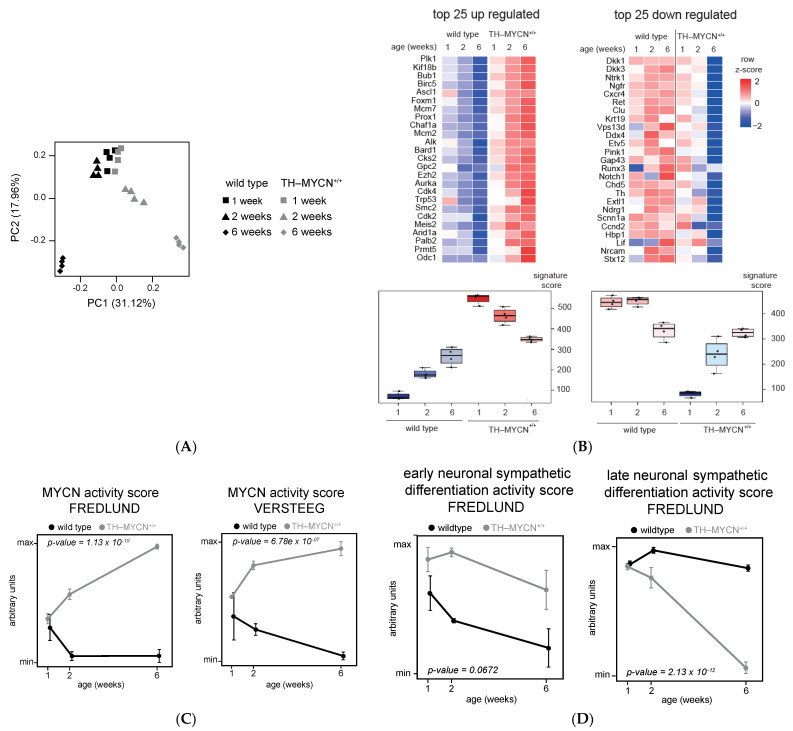
Time-resolved transcriptome data of murine TH-MYCN driven neuroblastoma tumors recapitulate established gene signatures. (**A**) Principal component analysis visualizing the variation in the dataset with PC1 discriminating the genotypes and PC2 the age of mice. (**B**) Heatmap showing the expression of 50 genes out of a total of 140 genes, previously reported as oncogenes, codrivers or tumor suppressor genes, that are the strongest up- (left) or downregulated (right) in TH-MYCN^+/+^ compared to wild type upon 6 weeks after birth. Each instance in the heatmap represents mean expression of 4 samples. Heat color reflects row-wise *z*-score. (**C**) Signature score analysis of MYCN activity in the hyperplasia dataset. Data represent mean signature score ± standard deviation of 4 samples. *y*-axis represents the ranks. *Gray*: TH-MYCN^+/+^ samples; *black*: wild type samples. *p*-value corresponds to the interaction term of a two-way ANOVA analysis. (**D**) Signature score analysis of early and late neuronal sympathetic differentiation in the hyperplasia dataset. Data represent mean signature score ± standard deviation of 4 samples. *y*-axis represents the ranks. *Gray*: TH-MYCN^+/+^ samples; *black*: wild-type samples. *p*-value corresponds to the interaction term of a two-way ANOVA analysis. (**E**) Signature score analysis of adrenergic and mesenchymal differentiation in the hyperplasia dataset. Data represent mean signature score ± standard deviation of 4 samples. *y*-axis represents the ranks. *Gray*: TH-MYCN^+/+^ samples; *black*: wild-type samples. *p*-value corresponds to the interaction term of a two-way ANOVA analysis. (**F**) Heatmap showing the expression of differentially expressed *MYC*(N) interacting proteins in TH-MYCN^+/+^ compared to wild-type samples upon 6 weeks of birth. Each instance in the heatmap represents mean expression of 4 samples. Heatmap colors reflect the row-wise *z*-score.

**Figure 2 cancers-13-04783-f002:**
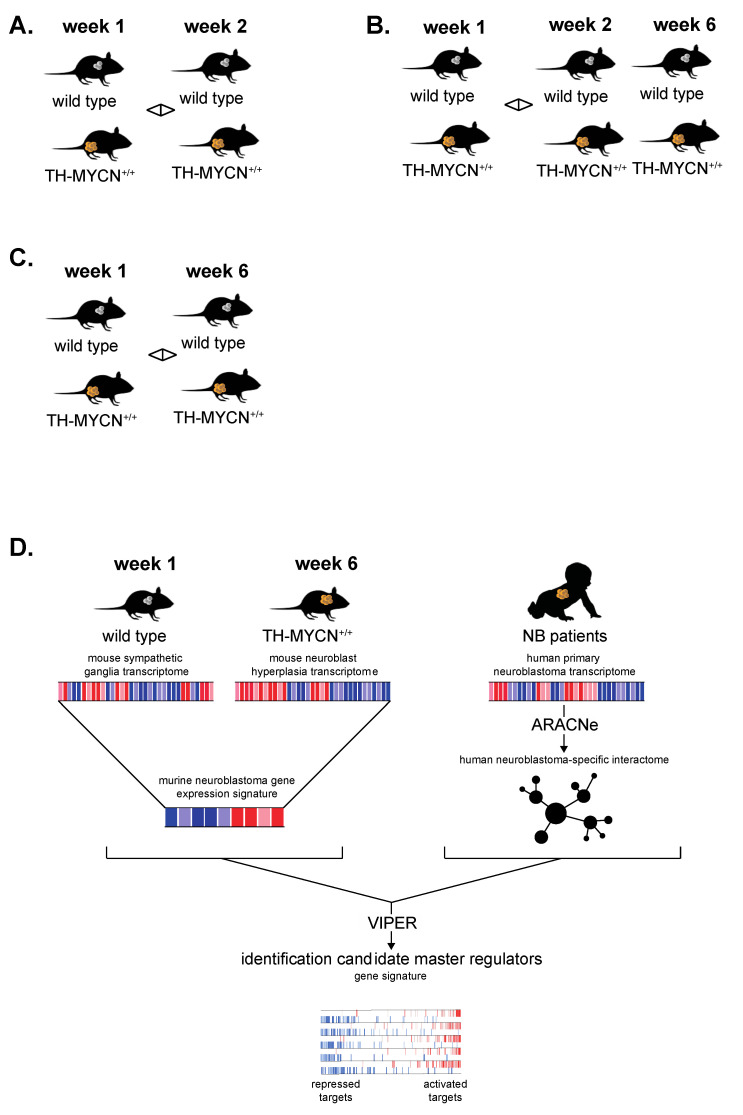
Graphical overview of the different transcriptome analyses performed in this study. Analysis (**A**) focuses on the identification of genes significantly differentially expressed between wild type and TH-MYCN^+/+^ from weeks 1 to 2 but not through week 6 after birth. Analysis (**B**) focuses on the identification of genes significantly differentially expressed between wild type and TH-MYCN^+/+^ from weeks 1 to 2 AND through week 6 after birth. Analysis (**C**) focuses on the identification of genes significantly differentially expressed between wild type and TH-MYCN^+/+^ from weeks 1 to 6 but not yet from weeks 1 to 2 after birth. Analysis (**D**) aims to identify putative master regulators of murine TH-MYCN-driven neuroblastoma development through cross-species transcriptome analysis of murine and human MYCN-driven neuroblastoma.

**Figure 3 cancers-13-04783-f003:**
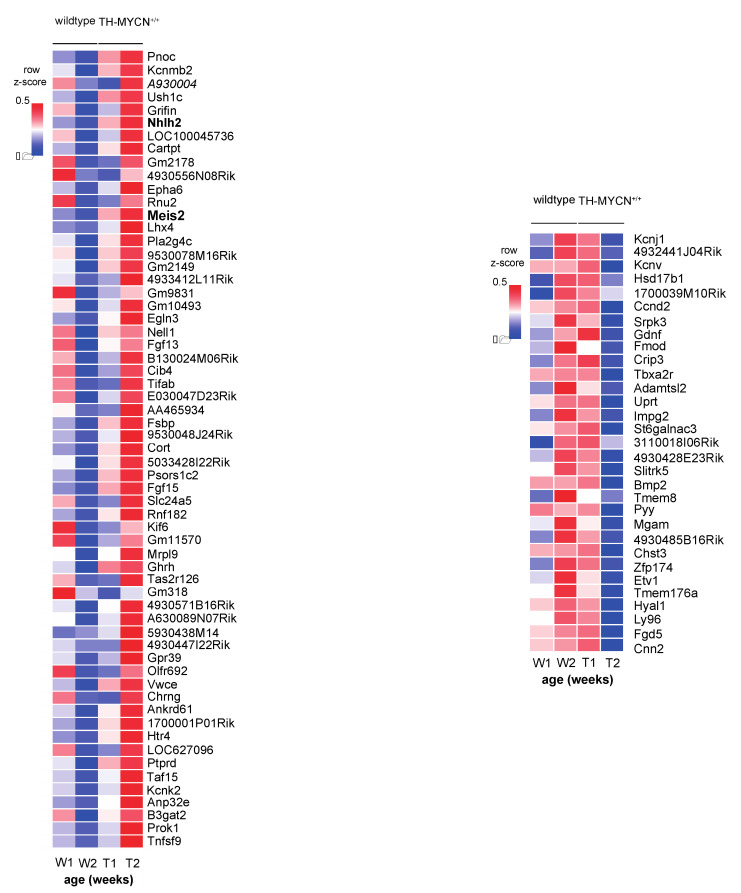
Identification of MEIS2 as a crucial factor of TH-MYCN driven neuroblastoma tumor initiation. (**A**) (top) Heatmaps displaying the up- (left, *p* < 0.01 and log_2_ fold change > 1) and downregulated genes (right, *p* < 0.01 and log_2_ fold change < −1) in TH-MYCN^+/+^ versus wild-type samples uniquely at 2 weeks versus week 1. (bottom) Boxplots (*p*-value corresponds to the interaction term of a two-way ANOVA analysis) represent the scoring of these gene lists as signatures in a large cohort of primary human neuroblastomas (GSE120572). (**B**) (left) Gene ontology analysis using Enrichr (maayanlab.cloud/Enrichr) based on the “Wiki pathway mouse” and “KEGG mouse” databases for pathway enrichment analysis for upregulated (*red*) and downregulated (blue) genes in TH-MYCN^+/+^ versus wild-type samples uniquely at 2 weeks of age; (right) network representation by means of the cytoscape plugin “Enrichment Map” of the Gene set enrichment analysis results using the C5_BP MSigDB collection for upregulated (*red*) genes in TH-MYCN^+/+^ versus wild-type samples uniquely at 2 weeks after birth. An FDR-q cut-off of 0.1 was used for network visualization. (**C**) Heatmaps depicting the genome-wide peak pattern of MEIS2, MYCN, H3K27ac, PHOX2B, GATA3, HAND2 and ISL1, all sorted according to the peak score of the MEIS2 binding sites in SK-N-BE(2)-C neuroblastoma cells. (**D**) Screenshots of the peak pattern in SK-N-BE(2)-C (left) and N206 (Kelly) (right) neuroblastoma cells of MYCN, MEIS2, H3K27ac, PHOX2B, GATA3, HAND2 and ISL1 at the *MEIS2* (top), *PHOX2B* (*middle*) and *RGS5* locus (bottom). (**E**) Log_2_ expression levels of *Isl1*, *Gata3*, *Phox2b*, *Hand2* (left) in comparison to *Meis2* and *Ascl1* (right) during murine TH-MYCN-driven neuroblastoma tumor development. Full lines represent expression in TH-MYCN^+/+^ transgenic mice, and dashed lines represent expression in wild-type mice. (**F**) (left) Single-cell analysis of *MEIS2* expression in the developing sympathoadrenal lineage points towards a strong expression in the sympathoblast population [37] (SCP: Schwann cell precursor); (right) single-cell analysis of *MEIS2* expression in different cell populations of the adrenal medulla. (**G**) Pharmacological MEIS2 inhibition using Meisi-2 resulted in a reduction of cell confluence of IMR32, N206 and SK-N-BE(2)-C neuroblastoma cells 72 h post exposure. AUC: area under the curve.

**Figure 4 cancers-13-04783-f004:**
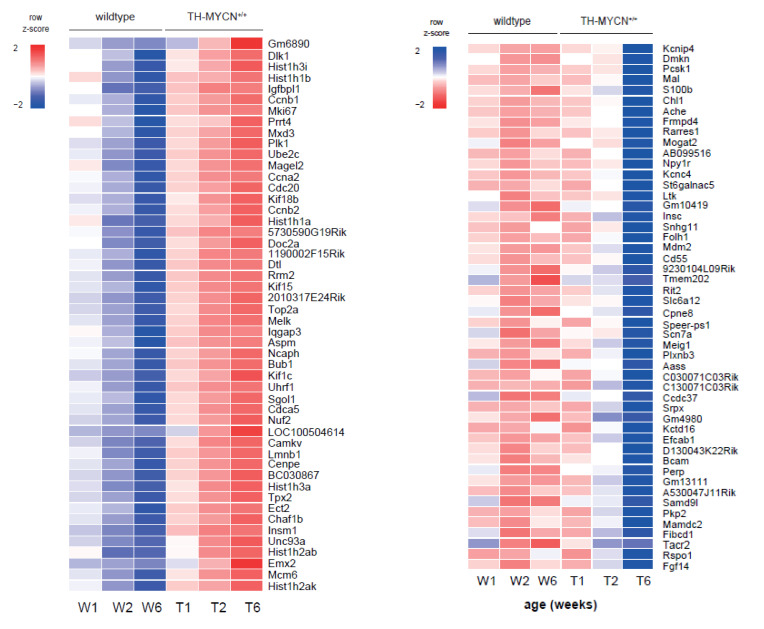
The KEOPS complex as putative key regulator of tumor maintenance in murine TH-MYCN driven neuroblastoma. (**A**) (top) Heatmaps displaying the top-100 up- (left, *p <* 0.01 and log_2_ fold change > 1) and downregulated genes (right, *p* < 0.01 and log_2_ fold change < −1) in TH-MYCN^+/+^ versus wild type samples common at 2 and 6 weeks of age versus week 1. (bottom) Boxplots (*p*-value corresponds to the interaction term of a two-way ANOVA analysis) represent the scoring of these gene lists as signatures in a large cohort of primary human neuroblastomas (GSE120572). (**B**) (left) Gene ontology analysis using Enrichr (maayanlab.cloud/Enrichr [24]) based on the “Wiki pathway mouse” and “KEGG mouse” databases for pathway enrichment analysis for up- (*red*) and downregulated (*blue*) genes in TH-MYCN^+/+^ versus wild-type samples overlapping between 2 and 6 weeks versus week 1; (right) network representation by means of the cytoscape plugin “Enrichment Map” of the Gene set enrichment analysis results using the C5_BP MSigDB collection for upregulated (*red*) genes in TH-MYCN^+/+^ versus wild-type samples overlapping between 2 and 6 weeks after birth. An FDR-q cut-off of 0.1 was used for network visualization. (**C**) (top) Heatmap displaying the top-100 up- (left, *p* < 0.01 and log_2_ fold change > 1) and downregulated genes (right, *p* < 0.01 and log_2_ fold change < −1) in TH-MYCN^+/+^ versus wild-type samples uniquely at 6 weeks of age versus week 1. (bottom) Boxplots (*p*-value corresponds to the interaction term of a two-way ANOVA analysis) represent the scoring of these gene lists as signatures in a large cohort of primary human neuroblastomas (GSE120572). (**D**) Gene ontology analysis using Enrichr (maayanlab.cloud/Enrichr [24]) based on the “Wiki pathway mouse” and “KEGG mouse” databases for pathway enrichment analysis for upregulated (*red*) genes in TH-MYCN^+/+^ versus wild type samples at uniquely 6 weeks of age versus week 1.

**Figure 5 cancers-13-04783-f005:**
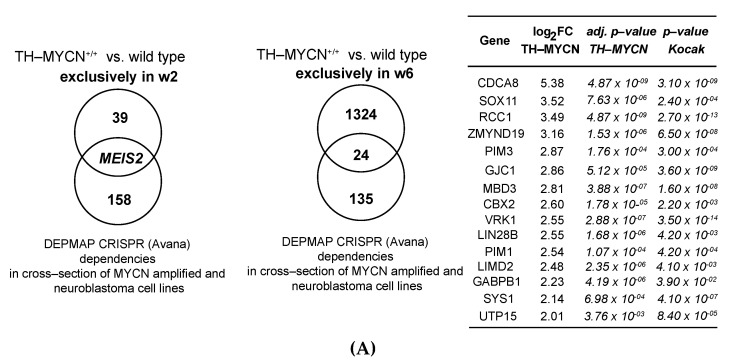
Identification of MYCN-driven dependencies during murine TH-MYCN driven neuroblastoma tumor formation. (**A**) (left) Overlap between upregulated genes in TH-MYCN^+/+^ versus wild-type samples upon 2 weeks after birth compared to Week 1, with 159 candidate genetic dependencies that were commonly found by the DEPMAP initiative (Avana CRISPR screen) as hits in MYCN-amplified and neuroblastoma cell lines and (right) Overlap between upregulated genes in TH-MYCN^+/+^ versus wild-type samples upon 6 weeks after birth compared to Week 1, with 159 candidate genetic dependencies that were commonly found by the DEPMAP initiative (Avana CRISPR screen) in MYCN-amplified and neuroblastoma cell lines (left), with an overview of 15 of the overlapping genes significantly correlated with poor overall patient survival in primary neuroblastoma tumors (GSE45547, *n* = 649) displaying their corresponding log_2_ fold change and (adjusted) *p*-value (right). (**B**) (left) Single-cell analysis of *CDCA8* expression in the developing sympathoadrenal lineage points towards nearly exclusive expression in proliferating sympathoblasts [49] (SCP: Schwann cell precursor); (right) single-cell analysis of *CDCA8* expression in different cell populations of the adrenal medulla [50]. (**C**) Kaplan–Meier analysis of overall patient survival based on gene expression levels of *CBX2*, *GJC1* and *LIMD2*, encoded on chromosome 17q in a large cohort of 649 primary neuroblastoma patients (GSE45547, *n* = 649) with high or low expression using median as a cut-off. *p*-values are a result of a log-rank test. (**D**) (top) Overlap between commonly upregulated genes in TH-MYCN^+/+^ versus wild-type samples upon 2 and 6 weeks after birth compared to Week 1, with 159 candidate genetic dependencies that were commonly found by the DEPMAP initiative (Avana CRISPR screen) as hits in MYCN-amplified and neuroblastoma cell lines (left), with an overview of 3 of the overlapping genes significantly correlated with poor overall patient survival in primary neuroblastoma tumors (GSE45547, *n* = 649) displaying their corresponding log_2_ fold change and (adjusted) *p*-value (right); (bottom) (left) Single-cell analysis of *CLSPN* expression in the developing sympathoadrenal lineage points towards nearly exclusive expression in (proliferating) sympathoblasts [49] (SCP: Schwann cell precursor); (right) Single-cell analysis of *CLSPN* expression in different cell populations of the adrenal medulla [50].

**Figure 6 cancers-13-04783-f006:**
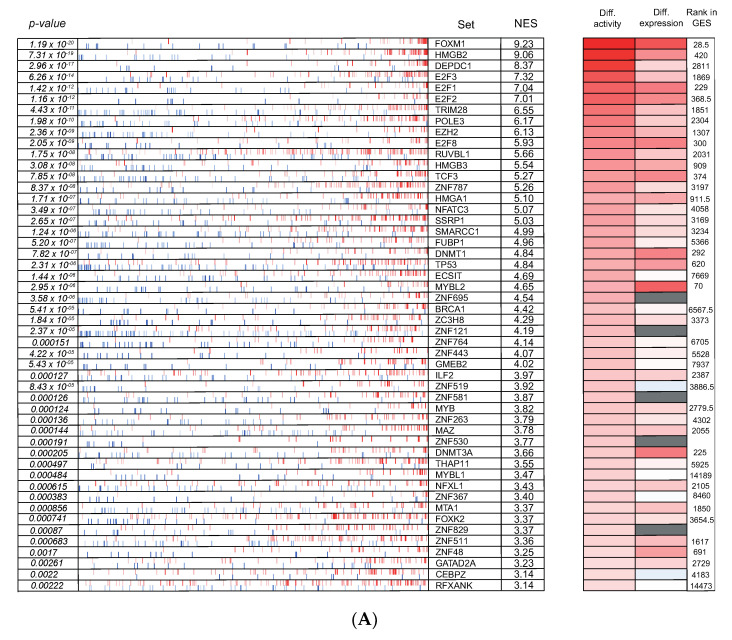
Identification of the DREAM complex and HMGB2 as key putative master regulators of TH-MYCN driven neuroblastoma tumorigenesis. (**A**) VIPER-inferred master regulators for neuroblastoma development. Each transcription factor regulon is indicated by a red and blue vertical bar, showing the distribution of activated (red) and repressed (blue) targets of each transcription factor ranked by differential expression in TH-MYCN^+/+^ versus wild-type mice upon 6 weeks after birth. The enrichment of each regulon on the neuroblastoma signature is indicated by the associated *p*-value (left) and normalized enrichment score (NES) (right). The heatmap on the right infers for each transcriptional regulator its differential activity and expression, as well as its rank in the gene expression signature. (**B**) (1) Heatmap representing the expression of the top-20 inferred master regulators after reranking according to differential expression from Weeks 1 to Week 6 after birth in TH-MYCN^+/+^ versus wild-type mice. (2) Signature score analysis of the DREAM complex targets in the hyperplasia dataset. Data represent mean signature score ± standard deviation of 4 samples. *y*-axis represents arbitrary units. *Gray*: TH-MYCN^+/+^ samples; *black*: wild type samples. *p*-value corresponds to the interaction term of a two-way ANOVA analysis. (3) (left) Single-cell analysis of *HMGB2* expression in the developing sympathoadrenal lineage points towards expression in the (proliferating) sympathoblasts and SCPs [49] (SCP: Schwann cell precursor); (right) Single-cell analysis of *HMGB2* expression in different cell populations of the adrenal medulla shows *HMGB2* expression mainly in the neuroblast and SCP population [50].

## Data Availability

In this study, several publicly available datasets were used: (1) gene expression profiles from 649 neuroblastoma tumors (GSE45547) to perform survival analysis, (2) gene expression profiles from 498 primary neuroblastomas (GSE49711) for master regulator analysis, (3) ChIP-seq data for GATA3, ISL1, HAND2, PHOX2B and H3K27ac in SK-N-BE(2)-C and N206 (Kelly) neuroblastoma cells (GSE94822) and (4) expression data from a cohort of primary medulloblastoma cases (GSE21140, =103). Microarray profiling results for all samples are available in the ArrayExpress database under Accession Number E-MTAB-3247. Furthermore, the processed data can be visualized via the R2 microarray Analysis and Visualization platform (http://r2.amc.nl, accessed on 22 September 2021) under experiment “Exp Nb Hyperplasia TH-MYCN—Ghent—24—de Preter—agmge8 × 60”. Furthermore, data can also be visualized through the Shiny application (https://shiny.dev.cmgg.be/app/01_hyperplasia_time_series, accessed on 22 September 2021) that we developed for this study. The raw ChIP-seq data files for all samples are available in the ArrayExpress database under Accession Number E-MTAB-10620.

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
