# Peer review of "MEIS2 Is an Adrenergic Core Regulatory Transcription Factor Involved in Early Initiation of TH-MYCN-Driven Neuroblastoma Formation"

_cancers, 2021, doi:10.3390/cancers13194783_

Round 1
Reviewer 1 Report
The authors have adequately addressed all of my concerns in the revised manuscript.
Reviewer 2 Report
The authors fully addressed all critical notes and suggestions, manuscript should be suitable for publication.
This manuscript is a resubmission of an earlier submission. The following is a list of the peer review reports and author responses from that submission.
Round 1
Reviewer 1 Report
De Wyn and colleagues present a temporal gene expression profiling study of MYCN-driven neuroblastoma development in the TH-MYCN mouse model of human MYCN-amplified neuroblastoma. The experimental design and strategy are logical, based on solid published studies. The study is focused on gene expression patterns during both early (hyperplasia in sympathetic ganglia) and late stages of neuroblastoma development in this model. Age-matched sympathetic ganglia from wild-type littermates are appropriate. Bioinformatic analyses are comprehensive. The data are presented in a systematic manner, followed by logical interpretations. The conclusions are generally supported by the results. Overall, this is a well-conducted study, generating interesting and potentially significant findings on genes and protein complexes involved in neuroblastoma pathogenesis. The report may also impact the neuroblastoma field by suggesting putative oncogenes and tumor suppressors for further functional, mechanistic, and translational studies.
Minor issues:
- Introduction should contain a brief description of the tumor development in TH-MYCN mice, including hyperplasia in early postnatal sympathetic ganglia and tumor formation at ~6 weeks in homozygous TH-MYCN mice with relevant citations. This information will help readers to understand the experimental design and strategy.
- Meisi-2 targets both Meis1 and Meis2. This should be mentioned when discussing the results. Also, the original reference for Meisi-2 should be cited (Sci Rep. 2020 May 14;10(1):7994).
- A brief overview of the concept of “cancer dependency genes” in Discussion may help readers to better understand and appreciate the significance of the results.
- The paper by Zha, Y., et al (Ref 29) presents evidence for MEIS2 as a transcriptional activator of FOXM1 (directly) and the DREAM complex (indirectly), which may suggest a mechanism for Meis2 in tumor development in this model. This should be discussed along with findings of this study.
- Line 281, should be a new paragraph after “(Figure 3G)”?
Reviewer 2 Report
The manuscript is focused on detailed inspection of the established neuroblastoma TH-MYCN mouse model in order to identify possible significant target driver genes common with original human tumors, which could be further used for medical verification. The transcriptome analysis reveals important role of MEIS2 and ASCL1 from detailed differential analysis of tumor model vs control in several time points and partially verified in human tumor materials. Additional global inspection of cross-species master regulators revealed a list of TFs involved e.g. FOXM1, DNMT1 etc. All the model result materials are available via additional online resources for direct inspection.
Indeed, the investigated TH-MYCN model is a quite useful research target, however it was derived from sympathetic ganglia, while recent studies, focusing on human single cell materials such as Kameneva et al Nat Genetics 2021 (ref31 in the manuscript) or Jansky et al Nat Genetics 2021 demonstrate cells-of-origin connection of neuroblastoma to other cell types. Therefore, important missing aspects in the manuscript are the global comparisons of the models to:
- human tumors in order to demonstrated how close is their correspondence e.g. using gene orthologs check if the models show closest similarity to MA neuroblastomas via correlation or unsupervised clustering in global bulk profiles
- latest published human single cell materials in order to confirm the suitability of the detected targets. It was partially started, but only one example was shown (Fig. 3F, MEIS2), while more global comparison would allow to verify evidence of more detected candidates.
There are some additional direct comments to the figures.
Figure 1A: As noted, only comparison within models is performed, but how do they reflect NB tumors?
Figure 1E: p-val for mesenchymal gene signature comparison is only 0.12 – any explanation of this?
Figure 1D: The selected genes are specific, showing figure as a boxplot from combination of genes would be more suitable to reflect the results
Figure 2: A bit hard to interpret this figure for blocks A,B,C. First, it’s unclear initially that they are connected to the top figure with models explained, maybe combining all with table borders would help. Also, it’s unclear what comparison blocks mean e.g. A is it week1 vs week2 only? Condition model vs wt should be stated e.g. in colors inside the boxes. Also, in all these comparisons might have overlaps in results. It would be nice to show general amounts of resulting genes and their intersections via venn diagram as a summary.
Figure 3A: If full results are shown, why the number of genes is so small in comparison A e.g. in Figure 4A/4C for B and C specifically only top 100 shown? This could be stated via summary e.g. Venn diagram as well. Also, in various manuscript sections different log2FC control used – at p7.l223 text abs log2FC is 2, while in the legend it is 1.
Figure 3C: why was the peak selection started from MYCN, but not MEIS2? What would be the effect if starting from MEIS2?
Figure 3D: Really hard to see anything, quality of figure is quite low. How to interpret the tracks: coverage/normalized? What are the y-axis limits per track? Are they different across gene loci or always the same? Locations of interest, e.g. promoters, probably should be marked.
Figure 3F : As noted, only 1 gene is checked, how about other discoveries? Also, additional possible dataset to inspect is from Jansky et al Nat Genetics 2021.
Figure 5: In stated MB cohorts only subsets of samples are MYCN associated. Could it be marked in the figure e.g. via shape? Possible way to distinguish them is simply to check MYCN expression. Also, if focusing only these MYCN-assoicated samples, does the high correlation remain?
Figure 7B, left side: which members are belonging to which complex block? How top10 are marked there (p19:l446), if the amount of genes is larger in the figure?
Methods section:
Settings/limits for analysis blocks are not provided e.g. p-val for limma are not stated here, should noted that specifically included in each figure legend
ChIP-seq data analysis is not described
There are no supplementary tables provided describing the results. For example, full GO comparison results might be useful for future inspection
